# Joint representation of molecular networks from multiple species improves gene classification

Christopher A. Mancuso[1], Kayla A. Johnson[2,3,4], Renming Liu[4], Arjun Krishnan[2,4]*

1 Department of Biostatistics and Informatics, University of Colorado Anschutz Medical Campus, Aurora, Colorado, United States of America, 2 Department of Biomedical Informatics, University of Colorado Anschutz Medical Campus, Aurora, Colorado, United States of America, 3 Department of Biochemistry and Molecular Biology, Michigan State University, East Lansing, Michigan, United States of America, 4 Department of Computational Mathematics, Science and Engineering, Michigan State University, East Lansing, Michigan, United States of America

* arjun.krishnan@cuanschutz.edu

**Data Availability Statement:** To enable the research community to work on these developments, we have made the relevant code

## Abstract

Network-based machine learning (ML) has the potential for predicting novel genes associated with nearly any health and disease context. However, this approach often uses network information from only the single species under consideration even though networks for most species are noisy and incomplete. While some recent methods have begun addressing this shortcoming by using networks from more than one species, they lack one or more key desirable properties: handling networks from more than two species simultaneously, incorporating many-to-many orthology information, or generating a network representation that is reusable across different types of and newly-defined prediction tasks. Here, we present GenePlexusZoo, a framework that casts molecular networks from multiple species into a single reusable feature space for network-based ML. We demonstrate that this multi-species network representation improves both gene classification within a single species and knowledge-transfer across species, even in cases where the inter-species correspondence is undetectable based on shared orthologous genes. Thus, GenePlexusZoo enables effectively leveraging the high evolutionary molecular, functional, and phenotypic conservation across species to discover novel genes associated with diverse biological contexts.

## Author summary

Our work addresses two major challenges; 1) computationally predicting the role a gene plays in various diseases, processes and phenotypes, and 2) accurately transferring genetic information discovered in one species to another. To simultaneously tackle both of these challenges, we developed the GenePlexusZoo method which builds a gene classification model by utilizing molecular interaction information from multiple species, seamlessly handling the complicated mapping of how genes across species are functionally related. We show that machine learning classifiers that utilize information from multiple species

from this study available at https://github.com/
krishnanlab/GenePlexusZoo-Mancusript and the
data available at https://zenodo.org/records/
10246207.

**Funding:** This work was primarily supported by US
National Institutes of Health (NIH) grant R35
GM128765 to AK and F32 GM134595 to CAM. The
funders had no role in study design, data collection
and analysis, decision to publish, or preparation of
the manuscript.

**Competing interests:** The authors have declared
that no competing interests exist.

outperform those that only consider information from a single species. Additionally, we
show that the GenePelxusZoo method is able to accurately transfer knowledge from one
species to another, even in the cases where no previous connection would have been
detected based solely on shared orthologous genes. Finally, we present an illustrative
example of how GenePlexusZoo can provide novel insights into a complicated genetic-
based disease.

## Introduction

Molecular interaction networks serve as genome-scale functional maps that help contextualize
and expand our knowledge about the various roles genes play in health and disease. Specifi-
cally, networks help in 'gene classification', the problem of predicting novel genes associated
with a biological context of interest—e.g., pathway, phenotype, or disease—based on a set of
known genes previously identified in multiple low-throughput studies or in single high-
throughput genetic screening or omics experiments. While network-based gene classification
has been shown to be a powerful approach [1–4], the gene/protein networks this approach
relies on are quite noisy and largely incomplete in many individual species including humans.
A potentially powerful way to address the shortcomings of individual networks is to leverage
the high evolutionary molecular, functional, and phenotypic conservation across species by
combining network information from multiple species to improve gene classification within a
single species.

Considering networks from multiple species also provides the opportunity to address
another major biological and biomedical challenge, which is transferring knowledge from
human to model species, or vice versa; a challenge that often causes clinical trials to fail [5–8].
Most current methods for transferring genetic and molecular knowledge across species are
restricted to using one-to-one orthologs, and not those with one-to-many and many-to-many
relationships. Further, deletion of orthologous pairs does not always give rise to the same phe-
notypes in different species [9], reinforcing the functional differences in the "same" genes
between model species and humans [10–12]. While methods are being developed for cross-
species knowledge transfer on the annotation level [13,14], or using expression data [15,16], it
is essential to develop methods that utilize genome-scale network information from multiple
species to build predictive models that capture the global functional context of nearly every
gene (even those uncharacterized) in all the species.

A few different approaches have been proposed to leverage network information across
multiple species. Some of these methods directly find similar functional modules in the net-
works of each species; either through graph alignment [17] or the use of gene families [18].
Methods such as MUNK [19], MUNDO [20], and ETNA [21] cast genes across pairs of species
(*e.g.*, human and mouse) into a joint low dimensional embedding space, anchoring the embed-
ding space together with one-to-one-orthologs. A recent deep learning approach, NetQuilt
[22], trains a neural network model using a multispecies network representation as input and
predicts the full set of gene ontology (GO) biological process [23] terms for each gene. There
also exist many methods that combine information from multiple networks within the same
species—using diffusion states [1], autoencoders [24] or graph convolutional networks [25]—
that could be readily adapted for the use in cross species knowledge transfer. The CAME
method integrates single-cell expression data across pairs of species using graph neural net-
works to improve cell-type assignment and discover shared cell-type-specific function in
homologous gene modules [26]. All of the above mentioned methods lack one or more

desirable properties: 1) the ability to handle networks from more than two species simultaneously, 2) incorporating many-to-many orthology information, or 3) generating a representation that is reusable across many tasks (*i.e.*, pathway, disease or phenotype prediction), including new tasks defined by users.

In this study, we explore the question of whether a joint representation of molecular networks from multiple species can be created so that this representation can be used on-the-fly for any gene classification task, both to improve predictions within a species and to prioritize novel genes across species, especially with the capability to seamlessly start with genes in any one species and prioritize genes in any other. To that end, we present GenePlexusZoo [**Fig 1**], a framework that casts molecular network information from multiple species (*i.e.*, human and five major research organisms) into a single reusable feature space for network-based machine learning (ML). We report three key results enabled by this framework on modeling cross-species gene associations. First, we show that, even when restricting training and prediction on gene annotations from a single target species, deriving the features of the ML classifier from multiple species outperforms ML classifiers with features from only the network from the target species. Second, we examine the case of training with annotations from one species and then predicting annotations in another species, and show that creating joint representations using node embedding with degree-weighted random walks allows for better knowledge transfer across species than more naive methods. Lastly, we present several examples of training a classifier using human disease-associated genes and predicting genes and biologically-

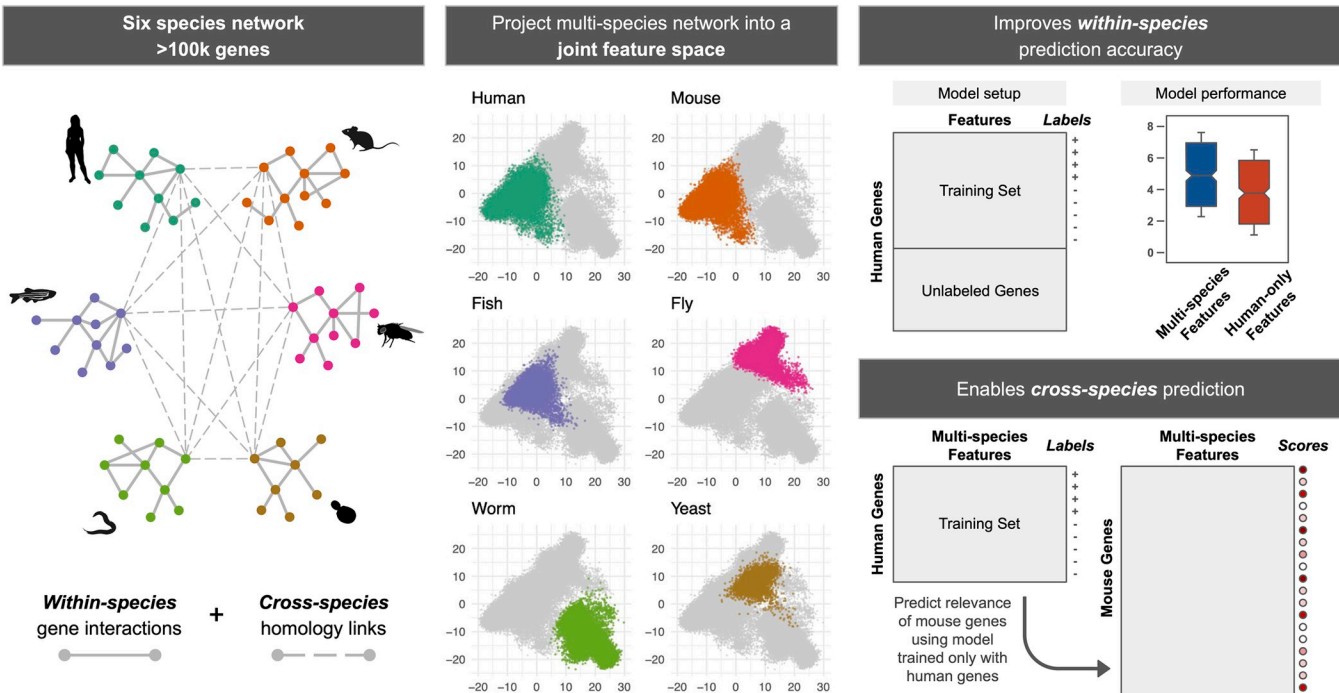

**Fig 1. Overview of GenePlexusZoo.** GenePlexusZoo is a framework for creating a joint representation of molecular networks from multiple species, which can in-turn be used for training machine learning (ML) classifiers for gene classification within and across species. The creation of the multi-species network representation entails: i) connecting gene networks from multiple species (in this study: human, mouse, fish, fly, worm, and yeast) so that pairs of genes belonging to the same orthologous group are connected across species (*cross-species* links; resulting in a multi-species network that can contain over 100k genes) and ii) projecting this network into a joint feature space by creating a low-dimensional embedding for each gene that captures its topological position and neighborhood. Here, we extensively test the ability of this joint network representation—and several other network-based feature sets containing the genes of one or more species—in predicting genes associated with numerous functions and phenotypes. Clipart images used to represent the different species were taken from PhyloPic (https://www.phylopic.org/images).

meaningful gene sets in research organisms, including multiple cases when the relevant gene sets (*e.g.*, pathways and phenotypes) in the other species share no common genes (orthologs) with the human disease. Overall, our work demonstrates that we can create effective joint representations of molecular networks from multiple species that can be easily reused for any within- and cross-species gene classification task. We propose GenePlexusZoo as a general framework where any previous [19–22], or future approaches for joint network embedding and learning can be adapted to take the place of our implementation to continue improving cross-species network-based gene classification.

## Methods

### Networks

**Processing networks.**   We chose networks from six species—*Homo sapiens* (human), *Mus musculus* (mouse), *Danio rerio* (fish), *Drosophila melanogaster* (fly), *Caenorhabditis elegans* (worm), and *Saccharomyces cerevisiae* (yeast)—from two databases, BioGRID [27] and IMP [28], that offer molecular networks that have genome-wide coverage [**Table A** in **S1 File**]. Bio-GRID is a sparse network consisting of binary edges (0 or 1) between genes based only on physical interactions between their protein products, whereas IMP is a dense network consisting of weighted edges (ranging continuously between 0 and 1) derived through computational integration of thousands of high-throughput genomic datasets. For each network, we used the Entrez IDs supplied by the databases to identify genes and, for each species, we removed any gene not present in that species "gene_info" file from the NCBI FTP server.

**Connecting networks across species.**   Retaining the within-species edges as they were, we added edges between cross-species gene pairs using orthology information from the eggNOG database [29], which contains pre-calculated orthologous groups for many species. For every pair of genes in a given orthologous group that come from different species, an edge was added between that pair of genes [**Fig A** in **S1 File**]. We examined two different ways to weight these cross-species edges [**Fig B** in **S1 File**]: first, giving all cross-species edges the same edge weight and, second, creating a directed edge with an edge weight that is a function of the within-species degree of the source node and how many total cross-species edges the source node has. See the **S1 File** for more information on the network properties and cross-species edge construction.

**Creating feature representations.**   Let $G = (V, E, W)$ denote an undirected molecular network, where $V$ is the set of vertices (genes from all species being considered), $E$ is the set of edges (relationships between genes), and $W$ is the set of edge weights (the strengths of the relationships). $G$ can be represented as a weighted adjacency matrix $A_{i,j} = W_{i,j}$, where $A \in R^{|V| \times |V|}$ and every row in this matrix can be treated as a feature vector for each gene (referred to as *AdjMat* in this work).

From $G$, we also derived a low-dimensional feature set through the process of node embedding. We used *PecanPy*, an optimized implementation of the *node2vec* (referred to as *N2V* in this work) algorithm [30,31]. At a high-level, *node2vec* generates a set of random walks from the network and then uses these walks as input into a neural network that places nodes that often occur close to each other in the walk close together in a low dimensional embedding space. In *node2vec*, the random walks are controlled using two parameters $p$ and $q$, in which a high value of $q$ keeps the walk local (a breadth-first search), and a high value of $p$ encourages outward exploration (a depth-first search). Additionally, the user can control the dimension of the embedding space ($d$), as well as the number and length of walks.

We additionally explored two more feature space representations (see **S1 File**): 1) a network transformation in which the edges of the network are smoothed using a random walk with

restart kernel (RWR), and 2) projecting the adjacency matrix representation into a lower-dimensional space using singular value decomposition (SVD). Various combinations of these methods and their corresponding hyperparameters were examined in this study (see **S1 File**).

## Gene set collections

**Gene set collections for different prediction tasks.**    To assess the performance of the ML classifiers, we investigated three different tasks: i) gene function prediction (using annotations from MyGene.info [32,33] mapped to terms in the Gene Ontology [23]), ii) gene phenotype prediction (using annotations from and terms defined in Monarch [34]), and iii) disease gene prediction (using annotations from DisGeNet [35] mapped to terms in the Disease Ontology [36]). For function and phenotype prediction, we were able to generate geneset collections for all six model species. Disease prediction was evaluated only for humans.

For each task, we created a gene set collection by first converting all annotations to Entrez gene IDs. For GO and DisGeNet, the annotations were then propagated from the more specific terms to the more general terms of the Gene and Disease Ontologies, respectively. Next, we filtered the genes to retain only those that were found in a given network and filtered the terms to only those that had between a specified minimum and maximum number of annotations. Mouse and human GO gene set collections were further filtered to only include terms in a previously curated set of specific terms [37]. Final sizes of all geneset collections used in this work are found in **Tables B-D** in **S1 File**.

**Deriving non-redundant gene sets.**    Next, we developed a procedure to take all gene sets within a collection and reduce them to a set of non-redundant terms. First, in a pairwise fashion, we calculated the set overlap and Jaccard scores of the genes annotated to each term. Then, we created a sparse term similarity graph by introducing edges between pairs of terms if the overlap and the Jaccard score were over predefined thresholds, and found connected components in this graph. If a connected component only had one term, that term was directly added to the non-redundant set. In the case where the component contained more than one term, we picked the term most similar to all others in the component as the representative term to be included in the non-redundant set.

**Evaluation scheme.**    Instead of splitting genes randomly into the training and testing sets, we devised a study-bias holdout scheme to evaluate the scenario that is close to the real-world situation of learning from well-characterized genes to predict novel un(der)-characterized genes. In this scheme, genes annotated to each term were split into training and testing sets based on how well-studied they were, with well-studied genes being considered training genes and more novel genes assigned to the test set. Here, we defined study-bias for each gene as the number of articles in PubMed in which that gene was referenced in, as determined in the *gene2pubmed* file (downloaded on 2020-09-23) from the NCBI Gene database [38]. After creating these gene splits, only terms with at least ten genes in both the training and testing sets were retained. For each term, negatives were chosen as any gene that had at least one annotation to another term in the set of all possible training and testing genes, where we considered genes neutral if they were from gene sets that highly overlapped (Fisher exact test with FDR < 0.05). In our previous work [4], we evaluated the GenePlexus approach on study-bias holdout as well as temporal holdout and 5-fold cross validation. However, due to the increased training time for the cross-species classifiers in this study (it can take up to 500 GBs of RAM and over 24 hours to train models for a single feature set–gene set collection–hyperparameter combination), we restricted ourselves to only study-bias holdout.

**Matched non-redundant gene sets across species.**    To evaluate GenePlexusZoo on its ability to transfer knowledge across species, we created gene set collections that contained only

matched terms across species. Due to an insufficient number of matched terms in Monarch phenotypes, this was only possible for the GO biological processes, where matched terms were generated pairwise between humans and each of the other model species. These gene sets were constructed by first converting genes to Entrez IDs and propagating the annotations from the more specific terms to more general terms in GO. The GO term annotations were then subset to the genes in the given network and only terms with at least ten training and testing genes were retained, based on the study-bias validation split. Gene annotations from both species were combined for each term and, using the procedure described above (based on a term similarity graph), we derived a non-redundant set of terms.

**Full gene set collections.**   For evaluation schemes where it is important to find the most relevant biological context, we use the full gene set collections instead of the non-redundant sets described above. For these analyses, we simply created gene set collections by converting gene annotations into Entrez IDs, propagating the annotations through the ontology (not done for Monarch), subsetting to the given network, and retaining terms with a given minimum-to-maximum number of annotations.

## Network-based gene classifiers

GenePlexus uses supervised-learning (SL) for network-based gene classification by using each gene's network neighborhoods as feature vectors, along with gene labels, in a classification algorithm. Here, we used logistic regression with L2 regularization with a strength of 1 as implemented by the *scikit-learn* [39] function *sklearn.linear_model.LogisticRegression*. For a more detailed explanation of how GenePlexus uses the SL classifier see [4,40,41]. Previous works in our group have extensively benchmarked the GenePlexus method using different shallow learning models [4] as well as graph neural networks [42].

**Evaluation metrics.**   In this study, we present results in terms of a modified area under the precision-recall curve. As each gene set can have a different number of positive examples (and, hence, different positive-to-negative proportions), to make classifier performance comparable across gene sets, we normalized the standard *auPRC* by the prior and *log* transform the ratio. Specifically, the metric presented throughout this study, $log_2(auPRC/prior)$, is given by the following set of equations:

$$auPRC = \sum_n (Recall_n - Recall_{n-1}) Precision_n$$

$$Precision = TP/(TP + FP)$$

$$Recall = TP/(TP + FN)$$

$$prior = \frac{P}{P+N} = \frac{TP + FN}{(TP + FN) + (FP + TN)}$$

where *TP* is true positives, *FN* is false negatives, *FP* is false positives, *TN* is true negatives, *P* and *N* being the number of positive and negative ground truth labels, respectively, and *n* is the $n^{th}$ threshold, where the number of thresholds is equal to the number of unique prediction probabilities from the classifier.

For unit changes in recall, the *auPRC* is a function of only the precision at various thresholds. If the classifier is random, the expected *precision* (fraction of all positive predictions that are true positives) at any threshold is a constant and is equal to the fraction of all labeled examples that are positives, i.e., the *prior*. Stated differently, if *x*% of all labeled genes are positives, then we also expect the same *x*% of a random subset of genes to be positives. Taking these two

facts into account, the expected *auPRC* of a random classifier equals the *prior*. Thus, $log_2(auPRC/prior)$ allows for the following interpretation: the number of 2-fold increases of the *auPRC* of the real classifier over the *auPRC* of a random classifier (in other words, the *auPRC* we would get by random chance; *e.g.*, a value of 1 indicates a 2-fold increase, a value of 2 indicates a 4-fold increase, etc.).

Precision-based metrics such as the *auPRC* are more suitable than the more popular area under the receiver-operating characteristic curve (*auROC*) for two reasons. First, gene classification is a highly imbalanced problem with many more negative examples than positive examples, and *auROC* is ill-suited for imbalanced problems [43]. Second, optimizing for precision controls for Type-1 error (false positives) [44], thus ensuring that the list of top predictions generated from a classifier contains as few false positives as possible. This goal of increasing precision is, therefore, more in line with the foremost goal of gene classification—to provide a list of top candidate genes with as few errors (false positives) as possible for further experimental follow-up—compared to increasing *auROC*, which ensures that, on average across the full genome-wide prediction ranking, positive examples are ranked higher than negative examples.

## Results

### Creating joint multi-species network representations using GenePlexusZoo

GenePlexusZoo was designed to build a robust reusable joint network-based feature representation for genes from multiple species for its machine learning (ML) classifier to improve gene classification both within and across species [**Fig 1**]. It begins with genes and their within-species relationships from human, mouse, fish, fly, worm, and yeast followed by combining them into a multi-species network by adding connections between pairs of genes across species based on orthology. Such a network built only using protein-protein interactions (PPI) from BioGRID contains 46,741 genes (across the six species) and nearly 1.59 million edges. The network built using predicted functional relationships from IMP contains 135,049 genes and nearly 566 million edges. Converting each of these large network objects into feature sets for ML entails making some choices that impact both gene classification performance and practical feasibility:

- Which species to include? For instance, which is better: the network from that species alone, a network combining that species and another well-studied species, or a network with all six species?

- How to connect genes across species? Should the new connections between cross-species gene pairs be weighted uniformly (with what weight?) or differently based on the degrees of the incident gene nodes?

- How to represent the network as features in the ML classifier? Can the adjacency matrix of the large multi-species network used as-is for ML or should the network be further smoothed (*e.g.*, using random walk with restart) or embedded into a lower dimension (*e.g.*, using singular value decomposition or node embedding).

To determine the best choices, we performed extensive evaluation for the human and mouse function and phenotype annotation tasks. For methods that directly consider network edge information, we tried AdjMat and RWR, and for network embedding methods, we considered N2V and SVD at different embedding dimensions (500, 2000, 10000). Many methods had additional hyperparameters that were tuned as well. Considering performance across all tasks as well as computational resources, we found that the adjacency matrix (AdjMat) is the best representation among those that directly consider the edges of the network as features.

*node2vec* (N2V) is the best-performing embedding method and has the additional properties of readily utilizing different weighting techniques and the ability to computationally handle combining networks from more than two species. For further discussion of the hyperparameter analysis, see **S1 File**.

## Multi-species network representations improve within-species gene classification

Once we had picked the best performing ways to create feature representations, we wanted to see if features derived from multiple species could offer improvement over features coming from a single species, even when the annotation task is restricted to only one species. For this analysis, we used non-redundant gene sets (see *Methods*) for function annotation (Gene Ontology) and phenotype annotation (Monarch) for the two networks considered in this study (BioGRID and IMP). We note here that the main goal of the work is to determine how changing feature representations affects classifier performance in a given task (i.e., network and gene set collection pair), and not to compare across tasks.

First, we examined the effect of adding multi-species network information on human gene annotation tasks. For functional annotation, using either BioGRID or IMP, features derived from pairs of species [**Fig 2**: blues] or features from all six species [**Fig 2**: orange] perform substantially better than features derived from a single species [**Fig 2**: greens]. For each network, we compared the log2(auPRC/prior) values each method that uses information from at least two species to the baseline method of Human-Only AdjMat using a Wilcoxon Signed Rank

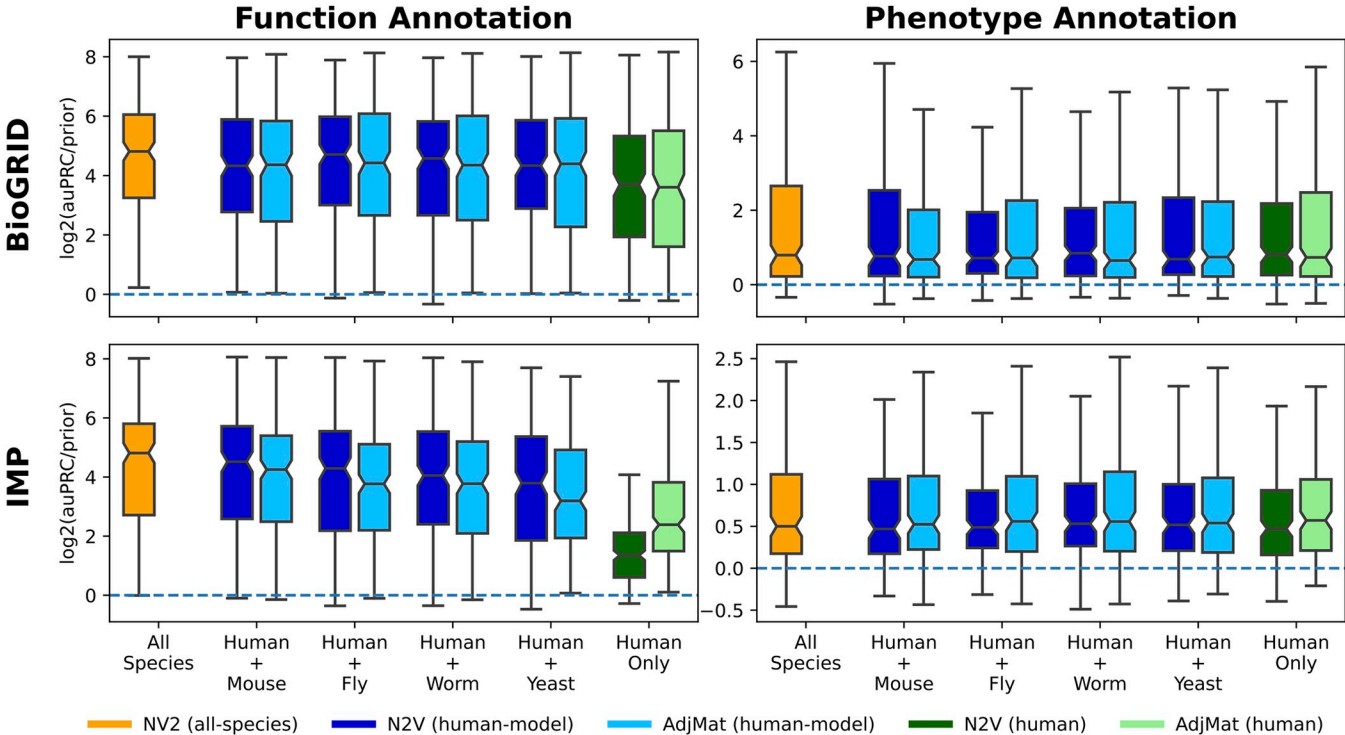

**Fig 2. Predicting function- and phenotype-associated human genes using human, human+model, and all-species network representations.** The boxplots show the $\log_2$(auPRC/prior) prediction performance of logistic regression classifiers trained to predict human gene annotation for two tasks: function (left column) and phenotype (right column). Features for the classifiers were created using all six species (orange), human plus one other species (blues), or human only (greens), and are represented using the adjacency matrix (AdjMat) or node embeddings (N2V). This analysis was performed using both the BioGRID (top row) and IMP (bottom row) networks.

Test, where a representation was considered significantly different from the baseline method is the Bonferroni adjusted p-value was below the family-side error rate (FWER) of 0.01 (see **Fig E** in **S1 File**). For BioGrid and IMP, every method was significantly better than Human-Only AdjMat, except for BioGRID's Human-Fish N2V representation. Phenotype annotation is a much harder task in general and all methods perform roughly the same no matter how the features are derived. For IMP, there are no significant differences, and for BioGRID the baseline method is better than N2V Human-Fish, and the AdjMAt representations for Human-Worm, Human-Fly and Human-Mouse.

Then, we examined the effect of adding multi-species network information on model-species gene annotation tasks, where we considered features only from the network for the species the annotation task pertains to [**Fig 3**: greens], adding human network information to the features [**Fig 3**: blues] and using features from all six species [**Fig 3**: orange]. For function annotation, the trends we observed were generally similar to that for human gene annotation–features derived from multiple species typically increase performance–albeit to a lesser extent. This can also be seen in the significance testing where 10 times a feature space derived from multiple species shows statistically significant improvement over the baseline method (here baseline method is the AdjMat representation using only information from the model species in which the prediction task is for, see **Fig F** in **S1 File**), whereas the baseline method never significantly outperforms a multi-species feature space. However, for model species annotation, the variation in performance using AdjMat or N2V increases substantially, especially for phenotype annotation. For BioGRID, the only statistically significant difference is the baseline method (AdjMat Yeast outperforming N2V All-Species). For IMP, the baseline method shows a significantly better performance 7 times, where the opposite is true only once (see **Fig F** in **S1 File**).

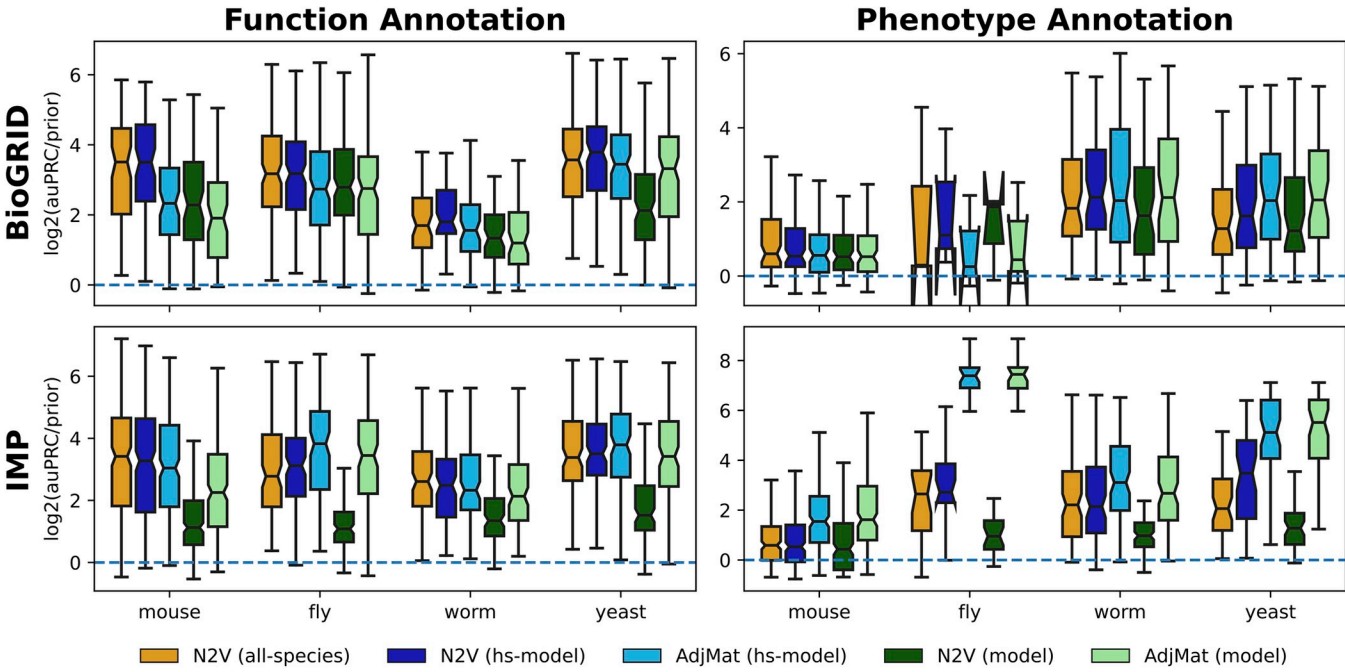

**Fig 3. Predicting function- and phenotype-associated model species genes using model, human+model, and all-species network representations.** The boxplots show the $\log_2$(auPRC/prior) prediction performance of logistic regression classifiers trained to predict model species gene annotations for two tasks: function (left column) and phenotype (right column). Features for the classifiers were created using networks from all six species (orange), human plus the other species (blue), or the model species only (green), and are represented using the adjacency matrix (AdjMat) or node embeddings (N2V). This analysis was performed using both the BioGRID (top row) and IMP (bottom row) networks.

We additionally checked the impact of explicitly marking orthologs of positive examples as additional positives when training the ML classifiers (on top of multi-species network implicitly containing this information through cross-species gene connections) [**Figs H–I** in **S1 File**]. We found that the addition of these potentially redundant positive examples results in occasional significant increases and decreases in performance for classifiers using the adjacency matrix representation, but consistently results in decreased performance for classifiers using the node embeddings.

As a final analysis of within-species gene classification, we compared results using a second orthology source, WORMHOLE, a machine learning based orthology prediction method that combines data across 17 distinct orthology databases for the six species used in this work [45]. In these results, we see that both orthology sources have comparable performance [**Figs J-K** in **S1 File**].

## Multi-species network embeddings enable cross-species gene classification

Next, we investigated the ability of these feature representations to transfer knowledge between human and model species. To that end, we first established a matched non-redundant gene set collection (see *Methods*) for each model species paired with human. For each human-model species pair, we trained a ML classifier using only the human annotations for a gene set in the matched gene set collection. Then, we generated prediction probabilities for every gene in the model species and calculated the log2(auPRC/prior) for every gene set in the matched gene set collection using model species annotations only. We performed this for three sets of features: AdjMat with features derived from human and model species, N2V with features derived from human and model species, and N2V with features derived from all six species.

We present the results of this analysis in terms of the ability of the human ML classifier to prioritize model species genes in the matched model-species gene set in which the classifier was trained [**Fig 4**; top row] as well as how the log2(auPRC/prior) of the model species annotations in the matched gene set is ranked compared to the log2(auPRC/prior) of the model species annotation in all the other (i.e. unmatched) gene sets in the matched gene set collection [**Fig 4**; bottom row]. These summaries clearly indicate that feature representations using node embedding are more accurate transferring knowledge from human to model species. These differences are all statistically significant except for Human-Worm N2V representation [**Fig G** in **S1 File**]. This trend also holds in the inverse case of transferring knowledge from model species to human (*e.g.*, training on model species gene sets and predicting the matched human gene sets) [**Fig L** in **S1 File**] for both network sources (IMP and BioGRID) [**Figs M–N** in **S1 File**], and nearly all but three cases are significantly different [**Fig G** in **S1 File**].

Delving further into the benefits of multi-species network embeddings (involving pairs of species or all species), we asked how does an ML classifier for a particular gene set perform in a given species when it is trained on gene annotations in that species compared to when it is trained on annotations from another species [**Figs O–R** in **S1 File**]. We observed that, for human gene annotation tasks, there is invariably a substantial drop in performance when the classifier is trained with model-species annotations compared to human annotations. However, the performances for the model species gene annotation tasks are variable. Compared to training on model-species annotations, training on human annotations results in reduced performance in fly and worm, but on-par performance in mouse and yeast, indicating that the knowledge in human is often a good proxy for these species.

As a final evaluation of cross-species transfer, we checked if training the ML classifier on both human and model species annotations would improve its performance in either species [**Figs S–V** in **S1 File**]. An analysis on prediction tasks in both human and model showed that,

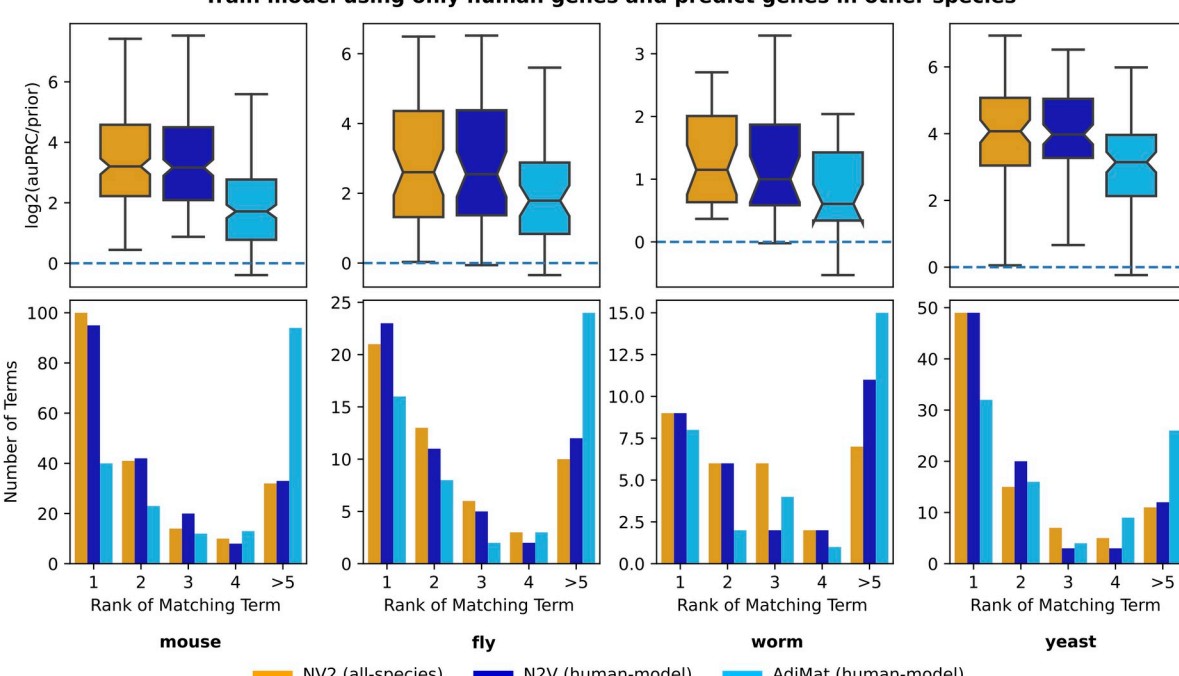

**Fig 4. Transferring knowledge from human to model species using human+model and all-species network representations.** The boxplots (top row) show the $\log_2$(auPRC/prior) prediction performance of logistic regression classifiers that were trained using only human genes annotated to a given biological process and then used to predict model species genes annotated to the same biological process. The barplots (bottom row) show the counts of where the $\log_2$(auPRC/prior) for the matched biological process ranks in comparison to the other processes in the collection. Features for the classifier were created pairwise between human and a model species with IMP networks using the AdjMat (light blue) and N2V (dark blue) representations, as well as features from all species using the N2V (orange) representation.

when using the multi-species network embeddings as features, including annotations from multiple species in the same classifier does not improve gene classification performance in a target species (human or model). The similarity of performance is likely because the information from different species is already inherent to the multi-species feature representation. This hypothesis is supported by the clear significant jump in performance between using features from a single species and features from multiple species.

## Mapping human diseases across species using all-species network embeddings

Having established the broad utility of all-species network embeddings in GenePlexusZoo, we explored the ability of this network representation to map human diseases across species and help discover meaningful and novel insights into diseases in all six model species. First, we identified ten human diseases that had top-performance in network-based disease-gene prediction (trained and tested only on human gene annotations and evaluated based on study-bias holdout for both BioGRID and IMP). Then, for each of these diseases, we trained a new ML classifier using all human genes annotated to that disease using the all-species network embeddings as features and made a prediction for every gene in all the five model species. For each model species, we could determine which processes and phenotypes were most enriched among the top-ranked "disease" genes in that species [**S1 Data**] (see **S1 File** for more information on how the enrichment scores are calculated).

To illustrate how GenePlexusZoo can provide insights across species, we focus on the ciliopathic disorder Bardet-Biedl Syndrome 1 (BBS) [46]. We choose this syndrome because it has

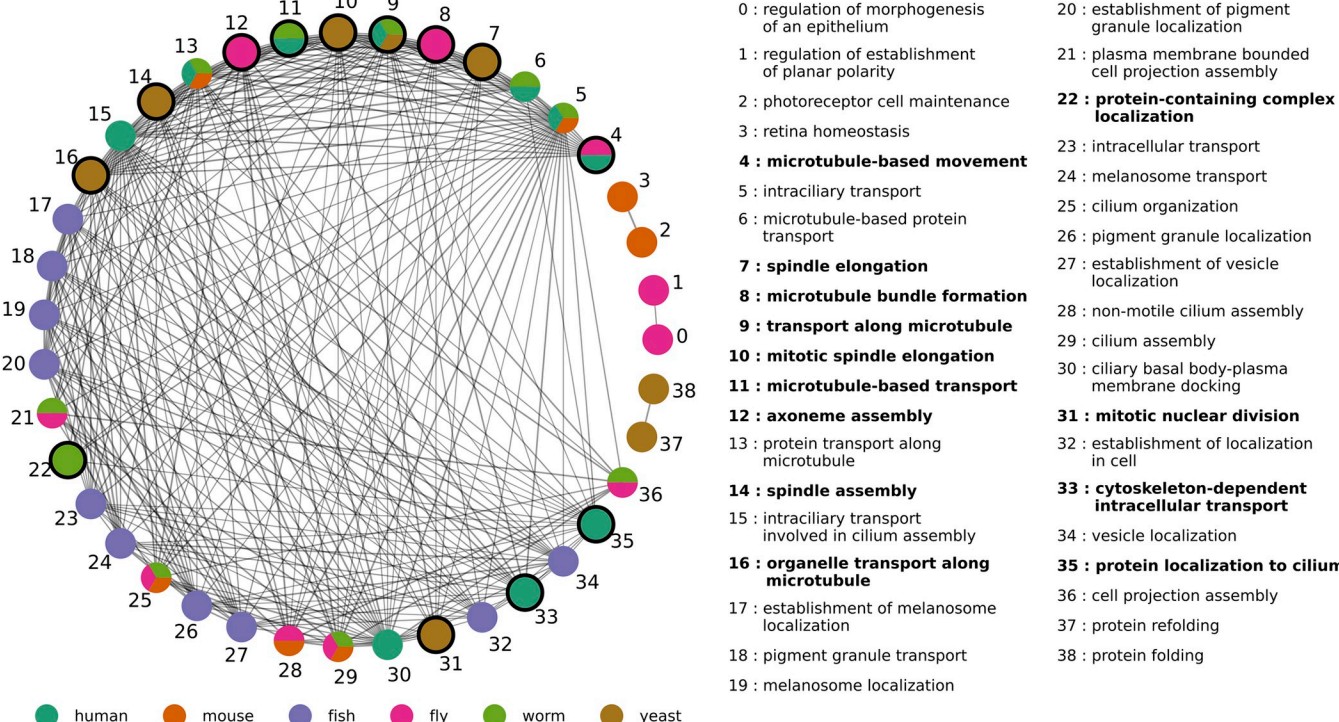

**Fig 5. The ten most enriched biological processes associated with the top genes in each species predicted to be related to Bardet-Biedl Syndrome 1 (BBS).** A classifier trained using human BBS genes and all-species N2V features was used to predict BBS-related genes in the five model species. The figure shows the ten most enriched biological processes associated with the top-ranked genes in each species. Nodes represent biological processes and are colored by which species they were identified in. Edges represent pairs of semantically similar processes (scaled Resnik similarity based on the Gene Ontology). Isolated nodes are not shown. Nodes with thick borders (with corresponding bolded labels in the legend) represent biological processes where, in at least one species, none of the annotated genes are orthologous to any human BBS gene.

a strong genetic component and impacts many processes across the body. Upon training an ML classifier only on human BBS genes and predicting genes in all model species, we observed that the top-ranked genes [**Table E** in **S1 File**], biological processes [**Table F** in **S1 File**], and phenotypes in the model species are relevant to cilium function and manifestations of the BBS disorder such as rod/cone dystrophy, polydactyly, and hypogonadism [**Table G** in **S1 File**]. To better highlight the relationships between the top biological processes across species, we summarized them as a graph [**Fig 5**] with nodes representing individual processes, node colors encoding the species the processes were identified in, and edges connecting semantically similar processes. Among these top processes were 13 where, in at least one species, the genes annotated to that biological process had no orthologs to any human gene annotated to BBS (see *Discussion* section for further discussion of this result).

## Embeddings bring together genes from multi-species networks

All the results above demonstrate the power of multi-species network representations in improving gene classification performance both within and across species, and in enabling knowledge transfer about human diseases to model species. These results also show that performance is influenced by how connections are made between genes across species in the multi-species network and how the multi-species network is turned into a feature representation (i.e., as is in the form of the adjacency matrix or as low-dimensional node embeddings; see *Methods* and **S1 File**). Therefore, we performed visual exploratory analysis to inspect the

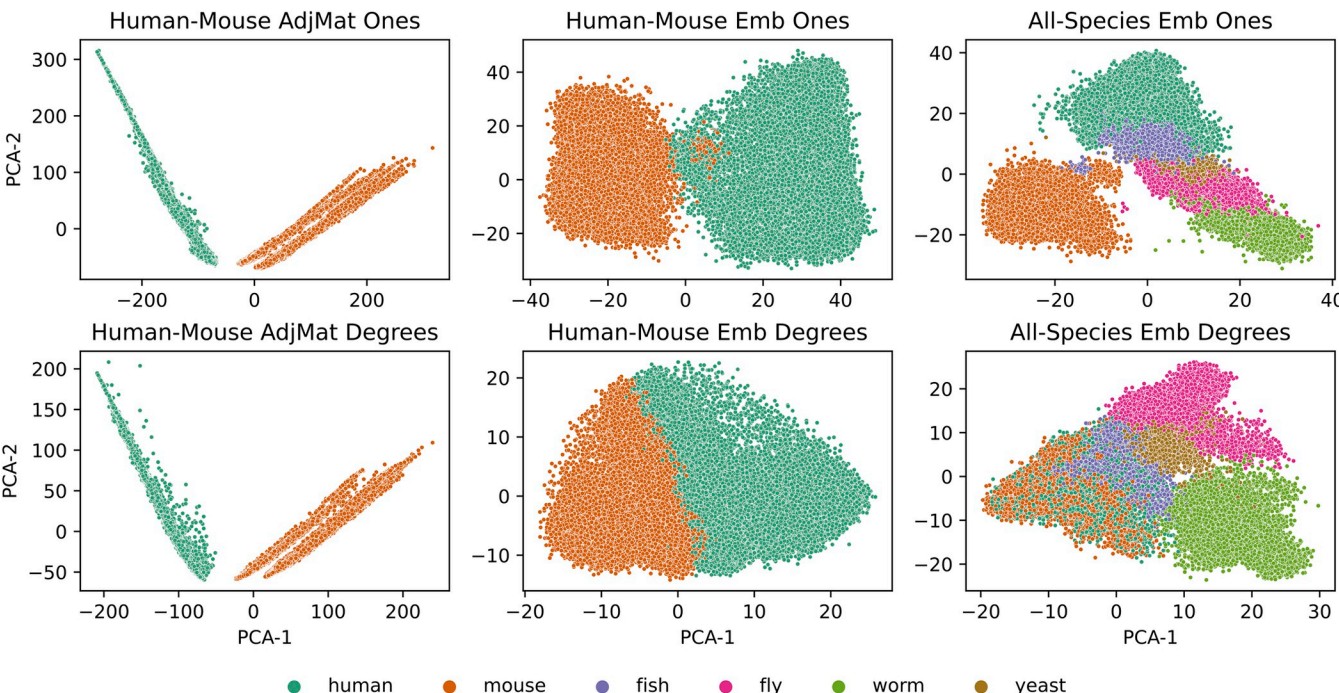

**Fig 6. PCA of adjacency matrix and node embedding representations.** Each PCA plot shows the top two PCA components from the human-mouse adjacency matrix (first column), the human-mouse node embedding (second column) and the all species node embedding (third column) representations. The cross-species edge weights were created by always using a uniform edge weight with a value of one (top row) or by using degree based edge weights (bottom row) for the IMP network.

feature spaces created through these different choices and to reason why the embeddings perform so well in cross-species tasks [**Fig 6**; **Figs X-Z** in **S1 File**].

The first observation from the plots comparing the features from different choices is that genes from different species are very disjoint in the adjacency matrix representation. This could perhaps explain why within species like fly and yeast, the adjacency matrix has such high performance, but for transferring knowledge across species, the adjacency matrix is the worst performing representation. On the other hand, multi-species network embeddings bring together genes from different species. Second, genes across species are brought much closer together when a degree-based weighting strategy is used to connect genes across species before embedding. These trends hold in pairwise settings (human and mouse [**Fig 6**]; human and other model species [**Figs X-Z** in **S1 File**]) and in the all-species setting.

## Discussion

GenePlexusZoo is a network-based gene classification framework that derives features for a supervised machine learning classifier from molecular networks from multiple species, providing a powerful foundation for improving gene classification accuracy within species and for transferring knowledge across species. GenePlexusZoo affords many ways to represent networks from multiple species including a novel method introduced here for generating a concise cross-species joint network representation that can be easily reused for any new gene classification task. This method involves connecting genes from different species together using orthology information and then jointly embedding them using a random-walk based algorithm, *node2vec* [30]. We find that the strategy for setting the weights between genes across

species can be very important and in particular we see that using a degree-based method that encourages walks to more readily traverse cross-species edges shows the best performance.

One of the goals of GenePlexusZoo is to offer a seamless way to handle one-to-many and many-to-many orthologs that goes beyond the typical workflows that use one-to-one orthologs to convert gene lists between human and model species. Previous computational approaches, such as MUNK [19], anchor networks together by "connecting" genes across species as one-to-one pairs. However, genes often act as coherent modules and simple one-to-one matching can be limiting to an analysis. It is often daunting and non-intuitive for researchers to handle one-to-many or many-to-many mappings, as a gene of interest in one species may be in a "functional orthologous group" with many genes from another species, thus clouding the transfer of knowledge. GenePlexusZoo inherently incorporates many-to-many gene mappings by generating network-based feature representations where genes from different species are close to each other if they are similar based on both sequence (orthology) and function (network neighborhood). Upon a classifier is trained on such a joint representation using annotations from one species, the resulting predictions in another species are returned as a ranked list, removing much ambiguity as to which of the one-to-many or many-to-many mapped genes are important for the user's task.

In addition to the problem of incorporating many-to-many orthologs, a challenge in generating cross species feature spaces lies in the increasing size of the total network space as more and more species are added. This restricts the use of naive methods such as concatenating adjacency matrices together, as those matrices quickly become computational intractable to work with. Graph neural networks designed for scalability, such as GraphSage with neighbor sampling [47] or GNNAutoScale [48], could be promising future directions to explore. However, currently these methods assume nodes across networks are equivalent, or at least highly overlapping. The *PecanPy* [31] implementation of the *node2vec* [30] node embedding method allows for a flexible way of incorporating connections across species, freedom in determining how walks are generated (i.e depth- or breadth-first searching), and an is optimized to handle networks with a large number of nodes, making it an ideal solution for GenePlexusZoo.

In this work, we performed extensive evaluations comparing our new multi-species network embedding with many other ways to represent networks from single and multiple species. Evaluation on single species annotation prediction tasks shows that the multi-species embedding almost always offers superior or equivalent performance to using the adjacency matrix. Notably however, for model species phenotype annotation prediction with the IMP network the adjacency matrix representation shows much better performance than embedding representations, especially in yeast and fly [**Fig 3**]. While the adjacency matrix representations show great performance for single species annotation prediction, the multi-species embedding representation could still be considered as these large feature representations that naively concatenate features from the different species appear to largely separate genes from different species [**Fig 6**], thus have trouble generalizing when transferring knowledge from one species to another [**Fig 4**] and require much greater computational resources to train. When creating low-dimensional representations using node embedding, building feature sets from more than one species almost always offers improvement over single species features. With the cross-species gene relationships already encoded in these representations, explicitly adding genes from multiple species as extra positives in the classifier does not improve performance [**Figs S-V** in **S1 File**].

After establishing the merits through the various evaluations, we applied GenePlexusZoo's multi-species network representation to a number of human disorders to explore the framework's ability to identify cross-species counterparts of human diseases and gain insights into the molecular and phenotypic basis of these conditions [**S1 Data**]. The case of ciliopathic disorder Bardet-Biedl Syndrome 1 (BBS) [46] provides an illustration of this application. Here,

we trained a classifier only using human BBS genes and predicted "BBS-associated" genes in the five model species. Many of the phenotypes associated with the top-ranked genes predicted by GenePlexusZoo in the model species correspond to the various manifestations of this syndrome including rod/cone dystrophy, polydactyly, central obesity, hypogonadism and kidney dysfunction [46]. In addition, most of the biological processes associated with the top-ranked genes speak to the ciliopathy of the disease, in particular a major component of BBS is disruption to intraflagellar transport [46,49,50]. This result indicates that the network-based classifier is general enough not to overfit to any one aspect of the syndrome and is able to recapitulate the diverse set of systems affected by BBS.

Further, this analysis has the potential to not only prioritize the best process/pathway to study in a given model organism, but also point to which model organism might be the best to study a given process/pathway. For example, ciliary behavior is related to the etiology of BBS and it has been studied extensively in fly [51,52]. Previous studies like this often guide the choice of model organism and, in this case, suggest fly as a good model. However, the association scores calculated using GenePlexusZoo for processes and, in particular, phenotypes related to ciliary behavior are quite low in fly [**S1 Data**]. On the other hand, the scores for these processes and phenotypes are much higher in worm, suggesting that it might be a better model than fly to study ciliary behavior. This hypothesis is supported by studies that show that Bardet-Biedl syndrome might be more driven by dysregulation of non-motile cilia, which is corroborated by GenePlexusZoo results [**Tables F-G** in **S1 File**] [53,54]. Further, many more genes have been associated with non-motile cilia in worm (32) compared to fly (13), possibly due to the fact that worms only have non-motile cilia in one cell type (sensory neuron) [55], thus lending itself as a better system for studying non-motile cilia diseases.

The application of GenePlexusZoo to BBS also brought to light a fundamental challenge in identifying 'phenologs', phenotypes in different species that are 'orthologous' by virtue of having a similar genetic and molecular basis (*i.e.*, similar functional modules composed of orthologous genes) [56]. While phenologs are critical for studying cross-species models of human diseases, finding them solely based on shared (orthologous) genes is problematic because our knowledge about the genes involved in both human diseases and model species phenotypes is far from complete. This incomplete knowledge can lead to bona fide phenologs being missed (false negatives) because the genes experimentally characterized thus far for the human and model phenotype happen to be non-overlapping (or minimally overlapping) even if they are part of the same underlying functional module in the two species. This challenge can be deftly remedied by using a network-based approach like GenePlexusZoo because it can relate disjoint sets of genes across species if they are close to each other in the underlying networks.

As evidence of this capability, GenePlexusZoo picks out highly similar processes and phenotypes in model species even when they share no orthologs with BBS in human [**Fig 5**]. A major mechanism underlying BBS is disruption to the meditation and regulation of microtubule-based intracellular transport processes [46,49,50]. A number of top predicted biological processes are microtubule-based intracellular transport processes (microtubule-based movement, microtubule-based movement, transport along microtubule, axoneme assembly, organelle transport along microtubule, etc.), however the model species genes annotated to those specific gene ontology terms have no known orthologs to the human BBS genes.

Another example is that in the case of organic acidemia (OA) [**Fig W** in **S1 File**], a class of inherited metabolic disorders that arise due to defects in intermediary metabolic pathways of carbohydrate, amino acids, and fatty acid oxidation [57]. Here, we observe that fatty acid biosynthetic process (FABP), a biological process highly relevant to OA [58], is top-ranked in mouse, but the two sets of genes (annotated to human OA and mouse FABP) have no shared orthologs.

Considering the similarities between all the top processes prioritized across species sheds additional light on the diversity of the molecular phenomena related to a particular human disease in terms of their biological relationships and the extent of our knowledge about them. For instance, the top processes for BBS in human and model species are highly similar to each other, with many of them being exactly the same across species [**Fig 5**, multicolored nodes]. On the other hand, in OA, there are zero biological processes that are exactly overlapping between species. Moreover, the top human biological processes have strong connections within themselves and not with biological processes from other species. The extreme case of this scenario can be seen in zebrafish in which the processes are only connected to each other and have no ortholog overlap with the human gene annotations for OA. Depending on the research question, these observations from the two human diseases imply that it might be harder to transfer research results across species for OA than BBS, or that, conversely, there are many more novel research avenues for studying OA in model species. Though we only highlight BBS and OA here, GenePlexusZoo identified biologically-meaningful processes and phenotypes across all the diseases we analyzed, with every case including model counterparts that have no gene overlap with the human disease [**S2 Data**].

Thus, GenePlexusZoo is a framework that will enable researchers to take any gene set they are interested in and build classifiers on-the-fly that leverage the molecular, functional, and phenotypic conservation across multiple species. The joint multi-species network representation of GenePlexusZoo simultaneously results in higher predictive power and offers a unique way to transfer knowledge directly across species compared to using a single species network. We note that the novelty of this work lies in the framework of this approach, and not so much on the exact manner in which the reusable network-based feature space is generated. Our implementation of this framework allows future work to explore how other similar methods for joint network representation could be adapted to continue improving cross-species knowledge transfer.

## Supporting information

**S1 Data. Processes and phenotypes in each model species that are most enriched among the top-ranked "human disease" genes in that species.** In each species, the top-ranked genes for each human disease were identified by training a classifier trained on GenePlexusZoo's multi-species network representation using the human disease genes and applying the trained model to predict genes in model species. The file contains a table of the Gene Ontology (GO) biological process genesets and Monarch phenotype genesets most enriched among these top-ranked genes. The columns of the file refer to the molecular network used to create the multi-species network representation (Network), the model species where predictions were made (Species), the Disease Ontology identifier of the human disease (DOID), the name of the disease (Disease Name), the number of human genes annotated to the disease (Number of Disease Genes), the subset of disease genes that have an ortholog in model species (Number of Disease Genes With Any Orthologs in Model Species), the GO or Monarch enrichment task (Task), the identifier of the process/phenotype (Term ID), the name of the process/phenotype (Term Name), the False Discovery Rate of the enrichment (FDR of Term in Model), the FDR-based enrichment rank (Rank), the number of genes in the model species annotated to that process/phenotype term (Number of Term Genes), the number of process/phenotype genes that have human orthologs (Number of Term Genes With Any Orthologs in Human), and the size of the gene overlap between the disease and the process/phenotype (Disease-Term Gene Overlap).
(TSV)

**S2 Data. The subset of processes and phenotypes in each model species that are most enriched among the top-ranked "human disease" genes in that species and have no gene overlap with the human disease.** This table contains a subset of rows in **S2 Data** where the size of the gene overlap between the disease and the process/phenotype (Disease-Term Gene Overlap) is zero. The columns are identical across the two tables.
(TSV)

**S1 File. Supplementary tables A–G, figures A–Z, and additional notes and references.** This information is organized under the following sections: multi-species networks, creating multi-species network representations, properties of gene set collections, optimal choices for gene classification using multi-species networks, statistical tests for main results, adding orthologs as positives, cross-species knowledge transfer, same-species prediction vs. full cross-species prediction, adding genes from multiple species during training, predicting model species counterparts of human diseases, and additional PCA plots.
(PDF)

## Acknowledgments

The authors wish to thank the members of the Krishnan Lab for valuable discussions. We additionally acknowledge Michigan State University for providing access to their high-performance computing cluster on which the work was performed.

## Author Contributions

**Conceptualization:** Christopher A. Mancuso, Arjun Krishnan.

**Data curation:** Kayla A. Johnson.

**Formal analysis:** Christopher A. Mancuso.

**Funding acquisition:** Christopher A. Mancuso, Arjun Krishnan.

**Investigation:** Christopher A. Mancuso, Kayla A. Johnson.

**Methodology:** Christopher A. Mancuso, Renming Liu, Arjun Krishnan.

**Project administration:** Christopher A. Mancuso, Arjun Krishnan.

**Resources:** Christopher A. Mancuso, Kayla A. Johnson, Renming Liu.

**Software:** Christopher A. Mancuso, Renming Liu.

**Supervision:** Arjun Krishnan.

**Validation:** Christopher A. Mancuso, Kayla A. Johnson.

**Visualization:** Christopher A. Mancuso.

**Writing – original draft:** Christopher A. Mancuso, Kayla A. Johnson, Renming Liu, Arjun Krishnan.

**Writing – review & editing:** Christopher A. Mancuso, Kayla A. Johnson, Renming Liu, Arjun Krishnan.

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
