## [Decision Letter · Decision Letter 0]

11 Sep 2023

Dear Krishnan,

Thank you very much for submitting your manuscript "Joint representation of molecular networks from multiple species improves gene classification" for consideration at PLOS Computational Biology.

As with all papers reviewed by the journal, your manuscript was reviewed by members of the editorial board and by several independent reviewers. In light of the reviews (below this email), we would like to invite the resubmission of a significantly-revised version that takes into account the reviewers' comments.

We cannot make any decision about publication until we have seen the revised manuscript and your response to the reviewers' comments. Your revised manuscript is also likely to be sent to reviewers for further evaluation.

Sincerely,

Shihua Zhang

Academic Editor

PLOS Computational Biology

Pedro Mendes

Section Editor

PLOS Computational Biology

Reviewer's Responses to Questions

**Comments to the Authors:**

Reviewer #1: This is a very good article and can be accepted in its present form.

Congratulations to all the authors for a good study.

Reviewer #2: The abstract outlines the development of "GenePlexusZoo," a framework for integrating molecular networks from multiple species into one feature space for network-based ML. It suggests this multi-species approach can enhance gene classification and knowledge transfer across species, even when inter-species correspondence is less evident based on orthologous genes.

The focus on a multi-species network representation and the emphasis on reusability for various types of prediction could be useful for researchers working on molecular network analyses and gene classification. The context is well established, stating the current limitations of single-species network analyses and the need for a unified approach.

The introduction establishes the importance of molecular interaction networks and highlights the challenges of single-species network analyses. It further emphasizes the value of cross-species knowledge transfer, especially for improving clinical trial outcomes. The paper places its claims within the context of existing literature, citing numerous studies that discuss the potential and pitfalls of current approaches. The identification of gaps in current methods namely an inability to handle multiple species networks, lack of incorporation of many-to-many orthology, and reusability across various tasks leads to the proposed solution, "GenePlexusZoo."

There are three key results that are mentioned in the Introduction and revisited later in the Results section:

Multi-species classifiers used on a single target species outperform classifiers trained on a single species.

The embedding methods introduced improve cross-species knowledge transfer for classification.

An example with a human disease gene-trained classifier being used to predict genes in other organisms.

The methods section begins with network selection by describing multi-species networks from two sources (one sparse - BioGRID, one dense - IMP). It continues to describe network connection with two edge weighting methods. Feature extraction is discussed next, primarily with the adjacency matrix and node2vec. There are several other methods outlined in S3 and they should be mentioned here as well (RWR, SVD). Gene set collection is divided into gene function prediction and phenotype prediction.

For evaluation, the study-bias holdout scheme is innovative. It closely aligns with real-world challenges where novel genes are being discovered and existing genes are being better characterized. It's not clear if random training and testing was attempted or not from the text. A comment should be added here on whether or not standard splitting methods were tested.

The gene classifier makes use of a standard logistic regression with L2 regularization supervised-learning approach, backed by previous studies. This is a simple yet powerful method for classification tasks but it doesn't look like any other classifiers or regularization methods were evaluated. There should be some more justification in the text for the single method or at least an acknowledgment of future research avenues to improve performance with different classifiers.

The evaluation metric of choice is auPRC and is selected over auROC due to the imbalanced dataset (making it more suitable for gene classification tasks). I have a couple of small issues with this section. The machine learning terminology may not be immediately understood by readers so precision and recall should be defined to make it instantly understandable to people familiar with other error rate terminology. Also, the explanation of auPRC and log2 needs to explicitly state what is increasing 2-fold over random chance. Additionally, in the next paragraph, it is not clear what the stated goal is in terms of error control. Saying 'top predictions are as correct as possible' is too vague and should be correctly defined. Also in the S3 material, it would be nice to see some illustrative examples of the PRC curves with key tests rather than just having the summary in box plots; something to justify the use of the summary statistics.

In the Results section, the first key result presented is under 'Multi-species network representations improve within-species gene classification'. I have a problem with the way these results are described compared to the accompanying Figure 2. The text describes features from six species performing significantly better than features from a single species. The word significant has a specific statistical meaning and should not be used here. Additionally, the BioGRID results do not show a strong result and the boxplots of log2(auPRC/prior) have largely overlapping quantiles across all categories. The only standout is the Human Only N2V with IMP for Function Annotation. Additionally, the scales should be the same (this goes for all comparison plots in both the main text and the S3 material) as it is not immediately apparent that the Phenotype IMP plot shows much worse results than the other three.

In figure 3 I have similar issues with the BioGRID results. The IMP results for Functional Annotation also show AdjMat (model) being close to or better than other methods for different organisms. There are some graphical artifacts with some of the box plots on the right side (also throughout S3) that should be fixed. The phenotype with IMP results has AdjMat stand out with yeast and fly.

The second section, 'Multi-species network embedding enable cross-species gene classification' shows the capabilities of transfer learning. Figure 4 agrees with the description in the text and shows a noticeable but not major increase in log2(auPRC/prior).

The last result, 'Mapping human diseases across species using all-species network embeddings' shows a practical exploration of Bardet-Biedl Syndrome 1 (BBS) as a demonstration of GenePlexusZoo's potential application. This case study approach, where abstract methods are applied to concrete problems, effectively communicates the tool's real-world significance.

The Discussion section is well written and addresses many of the particular elements with Figure 3 and Figure 4 from the Results section that I talked about earlier. It also presents the core claims of the study again which I feel were reasonably supported by the results. The significance of these claims lies in the potential of the framework to bridge the knowledge gap between species and enhance the predictive power in gene classification. This approach is innovative, with its strength lying in consolidating information from multiple species. The authors seem self-aware of the strengths and limitations of the methods. They point out that the novelty lies in the framework and not the exact method of generating the feature space, suggesting room for future research. However, while they mention that there are multiple ways to represent networks, a clearer emphasis on the challenges and limitations in creating these representations would add depth to the discussion.

I briefly reviewed the code on GitHub and found it was very well written and organized. The supporting information was well put together and the figures for the main text were appropriately selected.

Overall I found that the manuscript was well written but several things need to be addressed. The comparative plots should have the same scale on the log2(auPRC/prior) axis in my opinion. It is the same metric and presumably, we should be evaluating the performance of each comparison using a common scale. Some of the wording on the results seems too strong compared to the figures and I think a more careful treatment of the results and claims would be appropriate.

Reviewer #3: The article presents an idea of leveraging genetic connections between species to aid in network-based gene classification tasks. The idea is interesting, and the proposed framework seems to produce better results than using information from the target species only. This might be due to incompleteness of information on any single species or other factors. The article is fairly well written, and can be accepted for publication with minor revision.

Remarks (page numbering starts from the title page):

-In the abstract (and later on p. 2 in the introduction), it is said that “some recent methods […] using networks from more than one species, they lack [the property of] handling networks from multiple species”. This sounds odd and somewhat contradictory.

-p. 2, second paragraph: says “human to model species to human, or vice versa”. The sentence immediately following, has “most current methods” and “often”. Maybe one should be deleted.

-In my opinion, the bullet points in the beginning of the “Results” section as well as Figure 1 could fit better in the introduction (the final choice is of course the authors’).

-The hyperparameter selection results are completely left to supplementary material. Perhaps some summary could be included in the main text. Moreover, it seemed that these results were not really presented in their entirety. That is, the effect of different hyperparameter selections are not visualised anywhere. It is also not clear, how the N2Vselected parameters are chosen. Is that the overall best paramteter combination out of those tested?

-In the section “Multi-species network embeddings enable cross-species gene classification” in the results, it is really difficult to understand what kind of data is used as training and ground truth data. Moreover, the results include “Ranks of matching terms”, but they could be explained better. At least I cannot figure out how these results are obtained.

**Have the authors made all data and (if applicable) computational code underlying the findings in their manuscript fully available?**

Reviewer #1: Yes

Reviewer #2: Yes

Reviewer #3: Yes

PLOS authors have the option to publish the peer review history of their article (what does this mean?). If published, this will include your full peer review and any attached files.

Reviewer #1: No

Reviewer #2: **Yes: **R Zach Aandahl

Reviewer #3: No
---

## [Decision Letter · Decision Letter 1]

27 Nov 2023

Dear Krishnan,

Thank you very much for submitting your manuscript "Joint representation of molecular networks from multiple species improves gene classification" for consideration at PLOS Computational Biology.

As with all papers reviewed by the journal, your manuscript was reviewed by members of the editorial board and by several independent reviewers. In light of the reviews (below this email), we would like to invite the resubmission of a significantly-revised version that takes into account the reviewers' comments.

We cannot make any decision about publication until we have seen the revised manuscript and your response to the reviewers' comments. Your revised manuscript is also likely to be sent to reviewers for further evaluation.

Sincerely,

Shihua Zhang

Academic Editor

PLOS Computational Biology

Pedro Mendes

Section Editor

PLOS Computational Biology

Reviewer's Responses to Questions

**Comments to the Authors:**

Reviewer #3: I am happy with the revised manuscript and the authors' responses. The manuscript can be accepted as such.

Reviewer #4: In this paper, the authors propose GenePlexusZoo, a framework to cast molecular networks from multiple species into a single reusable feature space and this framework improves the gene classification tasks. Although the paper is overall clearly written, we have several major concerns about the methodology. In our opinion, these concerns, if not addressed, will preclude GenePlexusZoo to be a useful data analysis tool.

Below please find our detailed comments:

1. In section “introduction”, the authors mentioned that “Most current methods for transferring genetic and molecular knowledge across species are restricted to using one-to-one orthologs, and not those with one-to-many and many-to-many relationships”. However, a recent published paper, “Cross-species cell-type assignment from single-cell RNA-seq data by a heterogeneous graph neural network” which modeled many-to-many and one-to-many orthologs across species to do cross--species cell-type assignment and cross-species gene module extraction tasks. I think this should be also accessed in introduction or discussion.

2. In section “Connecting networks across species”, the authors used orthology information from the eggNOG database, we are interested that whether change the orthology information e.g., using orthology information from BioMart, will influence the results. The impact of such disparity should be assessed and evaluated seriously.

3. In section “Evaluation scheme” the authors provided a study-bias holdout scheme that is close to the real-world situation, however the results of cross validation did not be included. The time and memory consumption of the computations should, at least be reported to demonstrate the reasonability of not conducting cross-validation. The computing platform also needs to be detailed.

4. In Fig 2 and Fig 3, the authors reported the performance of predicting function- and phenotype-associated human genes using human, human + model, and all-species network representations. It seems that in the Function Annotation tasks, N2V (all-species) outperforms others in BioGRID and IMP networks. However, in Phenotype Annotation tasks, the AdjMat (human) method, which used Human only information achieved superior performance. So, what is the advantages of N2V (all-species) in this case? This, at the minimal, should be carefully evaluated.

5. In section “Mapping human diseases across species using all-species network embeddings”, the authors mentioned that “we observed that the top-ranked genes [Table SM5 in File S3], biological processes [Table SM6 in File S3], and phenotypes in the model species are relevant to cilium function and manifestations of the BBS disorder such as rod/cone dystrophy, polydactyly, and hypogonadism [Tables SM7 in File S3]”. Also, in the Fig 5, the authors highlight the biological processes, e.g., microtubule-based movement, spindle elongation, axoneme assembly, the authors should, at the minimal, cite some literatures to validate these biological processes are relevant to cilium function and manifestations of the BBS disorder.

6. In section “Gene set collections for different prediction tasks”, the authors mentioned that “we filtered the genes to retain only those that were found in a given network and filtered the terms to only those that had between a specified minimum and maximum number of annotations.” The minimum and maximum number of specified annotations should be explicitly specified.

7. In Fig 4, the legend of three colors should be added.

**Have the authors made all data and (if applicable) computational code underlying the findings in their manuscript fully available?**

Reviewer #3: Yes

Reviewer #4: Yes

PLOS authors have the option to publish the peer review history of their article (what does this mean?). If published, this will include your full peer review and any attached files.

Reviewer #3: No

Reviewer #4: No
---

## [Decision Letter · Decision Letter 2]

12 Dec 2023

Dear Krishnan,

Thank you very much for submitting your manuscript "Joint representation of molecular networks from multiple species improves gene classification" for consideration at PLOS Computational Biology. As with all papers reviewed by the journal, your manuscript was reviewed by members of the editorial board and by several independent reviewers. The reviewers appreciated the attention to an important topic. Based on the reviews, we are likely to accept this manuscript for publication, providing that you modify the manuscript according to the review recommendations.

Sincerely,

Shihua Zhang

Academic Editor

PLOS Computational Biology

Pedro Mendes

Section Editor

PLOS Computational Biology

Reviewer's Responses to Questions

**Comments to the Authors:**

Reviewer #4: I'd like to thank the authors for the updates and revisions. However, we still have some concerns about the authors’ claim in the rebuttal letter, below please find our detailed comments:

1. In section “introduction”, the authors mentioned that “The CAME method integrates single-cell expression data across pairs of species using graph neural networks to improve cell-type assignment and discover shared cell-type-specific function in homologous gene modules (X. Liu et al., 2023). All of the above mentioned methods lack one or more desirable properties: 1) the ability to handle networks from more than two species simultaneously, 2) incorporating many-to-many orthology information, or 3) generating a representation that is reusable across many tasks (i.e., pathway, disease or phenotype prediction)”. However, the statement in (2) may be misleading to readers since the CAME method do incorporate many-to-many orthology information,

2. In section “Evaluation scheme” the authors provided a study-bias holdout scheme that is close to the real-world situation, however the results of cross validation did not be included. In the revised version, the authors mentioned that “Here, models can have over 80k features, and thus the logistic regression algorithms need significant resources (500 GB RAM and over 24hours) to train models in a single gene set collection”. The running time should be accurate, i.e. 25hours or 48 hours, to let the reader make an accurate judgment.

3. In section “Mapping human diseases across species using all-species network

embeddings”, the authors mentioned that “we observed that the top-ranked genes [Table SM5 in File S3], biological processes [Table SM6 in File S3], and phenotypes in the model species are relevant to cilium function and manifestations of the BBS disorder such as rod/cone dystrophy, polydactyly, and hypogonadism”. In the rebuttal letter, the authors mentioned that “our claim is not that the highlighted (bolded) processes in Figure 5 that are being referred to in this comment (“e.g., microtubule-based movement, spindle elongation, axoneme assembly”, etc) are more relevant to BBS than the other processes in the figure. Rather, the processes in bold represent biological processes where, in at least one species, none of the annotated genes are orthologous to any human BBS gene”. I wondered, since these the processes in bold represent biological processes where, in at least one species, none of the annotated genes are orthologous to any human BBS gene, how to verify the relevance of these processes in the context of BBS? Since the title of figure 5 is “The ten most enriched biological processes associated with the top genes in each species predicted to be related to Bardet-Biedl Syndrome 1 (BBS)”, I believe it is necessary to verify the relevance of these processes to BBS.

**Have the authors made all data and (if applicable) computational code underlying the findings in their manuscript fully available?**

Reviewer #4: None

PLOS authors have the option to publish the peer review history of their article (what does this mean?). If published, this will include your full peer review and any attached files.

Reviewer #4: No

Figure Files:

Data Requirements:

Reproducibility:

References:

---

## [Editor Report · Decision Letter 3]

20 Dec 2023

Dear Krishnan,

We are pleased to inform you that your manuscript 'Joint representation of molecular networks from multiple species improves gene classification' has been provisionally accepted for publication in PLOS Computational Biology.

Best regards,

Shihua Zhang

Academic Editor

PLOS Computational Biology

Pedro Mendes

Section Editor

PLOS Computational Biology

---

## [Editor Report · Acceptance letter]

4 Jan 2024

PCOMPBIOL-D-23-01096R3 

Joint representation of molecular networks from multiple species improves gene classification

Dear Dr Krishnan,

I am pleased to inform you that your manuscript has been formally accepted for publication in PLOS Computational Biology. Your manuscript is now with our production department and you will be notified of the publication date in due course.

With kind regards,

Anita Estes
